# Sequence to Sequence Learning with Neural Networks

**Ilya Sutskever**
Google
ilyasu@google.com

**Oriol Vinyals**
Google
vinyals@google.com

**Quoc V. Le**
Google
qvl@google.com

## Abstract

Deep Neural Networks (DNNs) are powerful models that have achieved excellent performance on difficult learning tasks. Although DNNs work well whenever large labeled training sets are available, they cannot be used to map sequences to sequences. In this paper, we present a general end-to-end approach to sequence learning that makes minimal assumptions on the sequence structure. Our method uses a multilayered Long Short-Term Memory (LSTM) to map the input sequence to a vector of a fixed dimensionality, and then another deep LSTM to decode the target sequence from the vector. Our main result is that on an English to French translation task from the WMT-14 dataset, the translations produced by the LSTM achieve a BLEU score of 34.8 on the entire test set, where the LSTM's BLEU score was penalized on out-of-vocabulary words. Additionally, the LSTM did not have difficulty on long sentences. For comparison, a phrase-based SMT system achieves a BLEU score of 33.3 on the same dataset. When we used the LSTM to rerank the 1000 hypotheses produced by the aforementioned SMT system, its BLEU score increases to 36.5, which is close to the previous state of the art. The LSTM also learned sensible phrase and sentence representations that are sensitive to word order and are relatively invariant to the active and the passive voice. Finally, we found that reversing the order of the words in all source sentences (but not target sentences) improved the LSTM's performance markedly, because doing so introduced many short term dependencies between the source and the target sentence which made the optimization problem easier.

## 1 Introduction

Deep Neural Networks (DNNs) are extremely powerful machine learning models that achieve excellent performance on difficult problems such as speech recognition [13, 7] and visual object recognition [19, 6, 21, 20]. DNNs are powerful because they can perform arbitrary parallel computation for a modest number of steps. A surprising example of the power of DNNs is their ability to sort $N$ $N$-bit numbers using only 2 hidden layers of quadratic size [27]. So, while neural networks are related to conventional statistical models, they learn an intricate computation. Furthermore, large DNNs can be trained with supervised backpropagation whenever the labeled training set has enough information to specify the network's parameters. Thus, if there exists a parameter setting of a large DNN that achieves good results (for example, because humans can solve the task very rapidly), supervised backpropagation will find these parameters and solve the problem.

Despite their flexibility and power, DNNs can only be applied to problems whose inputs and targets can be sensibly encoded with vectors of fixed dimensionality. It is a significant limitation, since many important problems are best expressed with sequences whose lengths are not known a-priori. For example, speech recognition and machine translation are sequential problems. Likewise, question answering can also be seen as mapping a sequence of words representing the question to a

sequence of words representing the answer. It is therefore clear that a domain-independent method that learns to map sequences to sequences would be useful.

Sequences pose a challenge for DNNs because they require that the dimensionality of the inputs and outputs is known and fixed. In this paper, we show that a straightforward application of the Long Short-Term Memory (LSTM) architecture [16] can solve general sequence to sequence problems. The idea is to use one LSTM to read the input sequence, one timestep at a time, to obtain large fixed-dimensional vector representation, and then to use another LSTM to extract the output sequence from that vector (fig. 1). The second LSTM is essentially a recurrent neural network language model [28, 23, 30] except that it is conditioned on the input sequence. The LSTM's ability to successfully learn on data with long range temporal dependencies makes it a natural choice for this application due to the considerable time lag between the inputs and their corresponding outputs (fig. 1).

There have been a number of related attempts to address the general sequence to sequence learning problem with neural networks. Our approach is closely related to Kalchbrenner and Blunsom [18] who were the first to map the entire input sentence to vector, and is very similar to Cho et al. [5]. Graves [10] introduced a novel differentiable attention mechanism that allows neural networks to focus on different parts of their input, and an elegant variant of this idea was successfully applied to machine translation by Bahdanau et al. [2]. The Connectionist Sequence Classification is another popular technique for mapping sequences to sequences with neural networks, although it assumes a monotonic alignment between the inputs and the outputs [11].

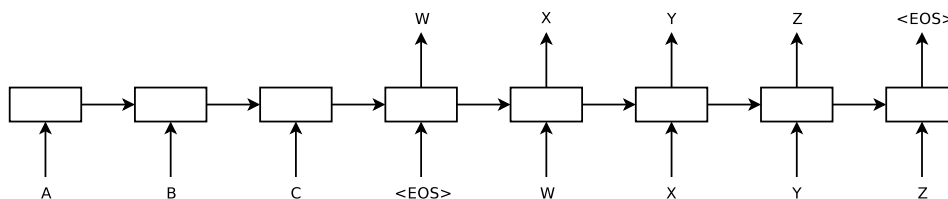

Figure 1: Our model reads an input sentence "ABC" and produces "WXYZ" as the output sentence. The model stops making predictions after outputting the end-of-sentence token. Note that the LSTM reads the input sentence in reverse, because doing so introduces many short term dependencies in the data that make the optimization problem much easier.

The main result of this work is the following. On the WMT'14 English to French translation task, we obtained a BLEU score of **34.81** by directly extracting translations from an ensemble of 5 deep LSTMs (with 380M parameters each) using a simple left-to-right beam-search decoder. This is by far the best result achieved by direct translation with large neural networks. For comparison, the BLEU score of a SMT baseline on this dataset is 33.30 [29]. The 34.81 BLEU score was achieved by an LSTM with a vocabulary of 80k words, so the score was penalized whenever the reference translation contained a word not covered by these 80k. This result shows that a relatively unoptimized neural network architecture which has much room for improvement outperforms a mature phrase-based SMT system.

Finally, we used the LSTM to rescore the publicly available 1000-best lists of the SMT baseline on the same task [29]. By doing so, we obtained a BLEU score of 36.5, which improves the baseline by 3.2 BLEU points and is close to the previous state-of-the-art (which is 37.0 [9]).

Surprisingly, the LSTM did not suffer on very long sentences, despite the recent experience of other researchers with related architectures [26]. We were able to do well on long sentences because we reversed the order of words in the source sentence but not the target sentences in the training and test set. By doing so, we introduced many short term dependencies that made the optimization problem much simpler (see sec. 2 and 3.3). As a result, SGD could learn LSTMs that had no trouble with long sentences. The simple trick of reversing the words in the source sentence is one of the key technical contributions of this work.

A useful property of the LSTM is that it learns to map an input sentence of variable length into a fixed-dimensional vector representation. Given that translations tend to be paraphrases of the source sentences, the translation objective encourages the LSTM to find sentence representations that capture their meaning, as sentences with similar meanings are close to each other while different

sentences meanings will be far. A qualitative evaluation supports this claim, showing that our model is aware of word order and is fairly invariant to the active and passive voice.

## 2    The model

The Recurrent Neural Network (RNN) [31, 28] is a natural generalization of feedforward neural networks to sequences. Given a sequence of inputs $(x_1, \ldots, x_T)$, a standard RNN computes a sequence of outputs $(y_1, \ldots, y_T)$ by iterating the following equation:

$$
\begin{aligned}
h_t &= \text{sigm}\left(W^{\text{hx}} x_t + W^{\text{hh}} h_{t-1}\right) \\
y_t &= W^{\text{yh}} h_t
\end{aligned}
$$

The RNN can easily map sequences to sequences whenever the alignment between the inputs the outputs is known ahead of time. However, it is not clear how to apply an RNN to problems whose input and the output sequences have different lengths with complicated and non-monotonic relationships.

A simple strategy for general sequence learning is to map the input sequence to a fixed-sized vector using one RNN, and then to map the vector to the target sequence with another RNN (this approach has also been taken by Cho et al. [5]). While it could work in principle since the RNN is provided with all the relevant information, it would be difficult to train the RNNs due to the resulting long term dependencies [14, 4] (figure 1) [16, 15]. However, the Long Short-Term Memory (LSTM) [16] is known to learn problems with long range temporal dependencies, so an LSTM may succeed in this setting.

The goal of the LSTM is to estimate the conditional probability $p(y_1, \ldots, y_{T'} | x_1, \ldots, x_T)$ where $(x_1, \ldots, x_T)$ is an input sequence and $y_1, \ldots, y_{T'}$ is its corresponding output sequence whose length $T'$ may differ from $T$. The LSTM computes this conditional probability by first obtaining the fixed-dimensional representation $v$ of the input sequence $(x_1, \ldots, x_T)$ given by the last hidden state of the LSTM, and then computing the probability of $y_1, \ldots, y_{T'}$ with a standard LSTM-LM formulation whose initial hidden state is set to the representation $v$ of $x_1, \ldots, x_T$:

$$
p(y_1, \ldots, y_{T'} | x_1, \ldots, x_T) = \prod_{t=1}^{T'} p(y_t | v, y_1, \ldots, y_{t-1}) \tag{1}
$$

In this equation, each $p(y_t | v, y_1, \ldots, y_{t-1})$ distribution is represented with a softmax over all the words in the vocabulary. We use the LSTM formulation from Graves [10]. Note that we require that each sentence ends with a special end-of-sentence symbol "<EOS>", which enables the model to define a distribution over sequences of all possible lengths. The overall scheme is outlined in figure 1, where the shown LSTM computes the representation of "A", "B", "C", "<EOS>" and then uses this representation to compute the probability of "W", "X", "Y", "Z", "<EOS>".

Our actual models differ from the above description in three important ways. First, we used two different LSTMs: one for the input sequence and another for the output sequence, because doing so increases the number model parameters at negligible computational cost and makes it natural to train the LSTM on multiple language pairs simultaneously [18]. Second, we found that deep LSTMs significantly outperformed shallow LSTMs, so we chose an LSTM with four layers. Third, we found it extremely valuable to reverse the order of the words of the input sentence. So for example, instead of mapping the sentence $a, b, c$ to the sentence $\alpha, \beta, \gamma$, the LSTM is asked to map $c, b, a$ to $\alpha, \beta, \gamma$, where $\alpha, \beta, \gamma$ is the translation of $a, b, c$. This way, $a$ is in close proximity to $\alpha$, $b$ is fairly close to $\beta$, and so on, a fact that makes it easy for SGD to "establish communication" between the input and the output. We found this simple data transformation to greatly boost the performance of the LSTM.

## 3    Experiments

We applied our method to the WMT'14 English to French MT task in two ways. We used it to directly translate the input sentence without using a reference SMT system and we it to rescore the n-best lists of an SMT baseline. We report the accuracy of these translation methods, present sample translations, and visualize the resulting sentence representation.

### 3.1 Dataset details

We used the WMT'14 English to French dataset. We trained our models on a subset of 12M sentences consisting of 348M French words and 304M English words, which is a clean "selected" subset from [29]. We chose this translation task and this specific training set subset because of the public availability of a tokenized training and test set together with 1000-best lists from the baseline SMT [29].

As typical neural language models rely on a vector representation for each word, we used a fixed vocabulary for both languages. We used 160,000 of the most frequent words for the source language and 80,000 of the most frequent words for the target language. Every out-of-vocabulary word was replaced with a special "UNK" token.

### 3.2 Decoding and Rescoring

The core of our experiments involved training a large deep LSTM on many sentence pairs. We trained it by maximizing the log probability of a correct translation $T$ given the source sentence $S$, so the training objective is

$$1/|\mathcal{S}| \sum_{(T,S) \in \mathcal{S}} \log p(T|S)$$

where $\mathcal{S}$ is the training set. Once training is complete, we produce translations by finding the most likely translation according to the LSTM:

$$\hat{T} = \arg \max_T p(T|S) \tag{2}$$

We search for the most likely translation using a simple left-to-right beam search decoder which maintains a small number $B$ of partial hypotheses, where a partial hypothesis is a prefix of some translation. At each timestep we extend each partial hypothesis in the beam with every possible word in the vocabulary. This greatly increases the number of the hypotheses so we discard all but the $B$ most likely hypotheses according to the model's log probability. As soon as the "<EOS>" symbol is appended to a hypothesis, it is removed from the beam and is added to the set of complete hypotheses. While this decoder is approximate, it is simple to implement. Interestingly, our system performs well even with a beam size of 1, and a beam of size 2 provides most of the benefits of beam search (Table 1).

We also used the LSTM to rescore the 1000-best lists produced by the baseline system [29]. To rescore an n-best list, we computed the log probability of every hypothesis with our LSTM and took an even average with their score and the LSTM's score.

### 3.3 Reversing the Source Sentences

While the LSTM is capable of solving problems with long term dependencies, we discovered that the LSTM learns much better when the source sentences are reversed (the target sentences are not reversed). By doing so, the LSTM's test perplexity dropped from 5.8 to 4.7, and the test BLEU scores of its decoded translations increased from 25.9 to 30.6.

While we do not have a complete explanation to this phenomenon, we believe that it is caused by the introduction of many short term dependencies to the dataset. Normally, when we concatenate a source sentence with a target sentence, each word in the source sentence is far from its corresponding word in the target sentence. As a result, the problem has a large "minimal time lag" [17]. By reversing the words in the source sentence, the average distance between corresponding words in the source and target language is unchanged. However, the first few words in the source language are now very close to the first few words in the target language, so the problem's minimal time lag is greatly reduced. Thus, backpropagation has an easier time "establishing communication" between the source sentence and the target sentence, which in turn results in substantially improved overall performance.

Initially, we believed that reversing the input sentences would only lead to more confident predictions in the early parts of the target sentence and to less confident predictions in the later parts. However, LSTMs trained on reversed source sentences did much better on long sentences than LSTMs

trained on the raw source sentences (see sec. 3.7), which suggests that reversing the input sentences results in LSTMs with better memory utilization.

## 3.4 Training details

We found that the LSTM models are fairly easy to train. We used deep LSTMs with 4 layers, with 1000 cells at each layer and 1000 dimensional word embeddings, with an input vocabulary of 160,000 and an output vocabulary of 80,000. We found deep LSTMs to significantly outperform shallow LSTMs, where each additional layer reduced perplexity by nearly 10%, possibly due to their much larger hidden state. We used a naive softmax over 80,000 words at each output. The resulting LSTM has 380M parameters of which 64M are pure recurrent connections (32M for the "encoder" LSTM and 32M for the "decoder" LSTM). The complete training details are given below:

- We initialized all of the LSTM's parameters with the uniform distribution between -0.08 and 0.08

- We used stochastic gradient descent without momentum, with a fixed learning rate of 0.7. After 5 epochs, we begun halving the learning rate every half epoch. We trained our models for a total of 7.5 epochs.

- We used batches of 128 sequences for the gradient and divided it the size of the batch (namely, 128).

- Although LSTMs tend to not suffer from the vanishing gradient problem, they can have exploding gradients. Thus we enforced a hard constraint on the norm of the gradient [10, 25] by scaling it when its norm exceeded a threshold. For each training batch, we compute $s = \|g\|_2$, where $g$ is the gradient divided by 128. If $s > 5$, we set $g = \frac{5g}{s}$.

- Different sentences have different lengths. Most sentences are short (e.g., length 20-30) but some sentences are long (e.g., length $> 100$), so a minibatch of 128 randomly chosen training sentences will have many short sentences and few long sentences, and as a result, much of the computation in the minibatch is wasted. To address this problem, we made sure that all sentences within a minibatch were roughly of the same length, which a 2x speedup.

## 3.5 Parallelization

A C++ implementation of deep LSTM with the configuration from the previous section on a single GPU processes a speed of approximately 1,700 words per second. This was too slow for our purposes, so we parallelized our model using an 8-GPU machine. Each layer of the LSTM was executed on a different GPU and communicated its activations to the next GPU (or layer) as soon as they were computed. Our models have 4 layers of LSTMs, each of which resides on a separate GPU. The remaining 4 GPUs were used to parallelize the softmax, so each GPU was responsible for multiplying by a $1000 \times 20000$ matrix. The resulting implementation achieved a speed of 6,300 (both English and French) words per second with a minibatch size of 128. Training took about a ten days with this implementation.

## 3.6 Experimental Results

We used the cased BLEU score [24] to evaluate the quality of our translations. We computed our BLEU scores using `multi-bleu.pl`[1] on the *tokenized* predictions and ground truth. This way of evaluating the BELU score is consistent with [5] and [2], and reproduces the 33.3 score of [29]. However, if we evaluate the state of the art system of [9] (whose predictions can be downloaded from `statmt.org\matrix`) in this manner, we get 37.0, which is greater than the 35.8 reported by `statmt.org\matrix`.

The results are presented in tables 1 and 2. Our best results are obtained with an ensemble of LSTMs that differ in their random initializations and in the random order of minibatches. While the decoded translations of the LSTM ensemble do not beat the state of the art, it is the first time that a pure neural translation system outperforms a phrase-based SMT baseline on a large MT task by

| Method | test BLEU score (ntst14) |
|---|---|
| Bahdanau et al. [2] | 28.45 |
| Baseline System [29] | 33.30 |
| Single forward LSTM, beam size 12 | 26.17 |
| Single reversed LSTM, beam size 12 | 30.59 |
| Ensemble of 5 reversed LSTMs, beam size 1 | 33.00 |
| Ensemble of 2 reversed LSTMs, beam size 12 | 33.27 |
| Ensemble of 5 reversed LSTMs, beam size 2 | 34.50 |
| Ensemble of 5 reversed LSTMs, beam size 12 | **34.81** |

Table 1: The performance of the LSTM on WMT'14 English to French test set (ntst14). Note that an ensemble of 5 LSTMs with a beam of size 2 is cheaper than of a single LSTM with a beam of size 12.

| Method | test BLEU score (ntst14) |
|---|---|
| Baseline System [29] | 33.30 |
| Cho et al. [5] | 34.54 |
| State of the art [9] | **37.0** |
| Rescoring the baseline 1000-best with a single forward LSTM | 35.61 |
| Rescoring the baseline 1000-best with a single reversed LSTM | 35.85 |
| Rescoring the baseline 1000-best with an ensemble of 5 reversed LSTMs | **36.5** |
| Oracle Rescoring of the Baseline 1000-best lists | ∼45 |

Table 2: Methods that use neural networks together with an SMT system on the WMT'14 English to French test set (ntst14).

a sizeable margin, despite its inability to handle out-of-vocabulary words. The LSTM is within 0.5 BLEU points of the previous state of the art by rescoring the 1000-best list of the baseline system.

### 3.7 Performance on long sentences

We were surprised to discover that the LSTM did well on long sentences, which is shown quantitatively in figure 3. Table 3 presents several examples of long sentences and their translations.

### 3.8 Model Analysis

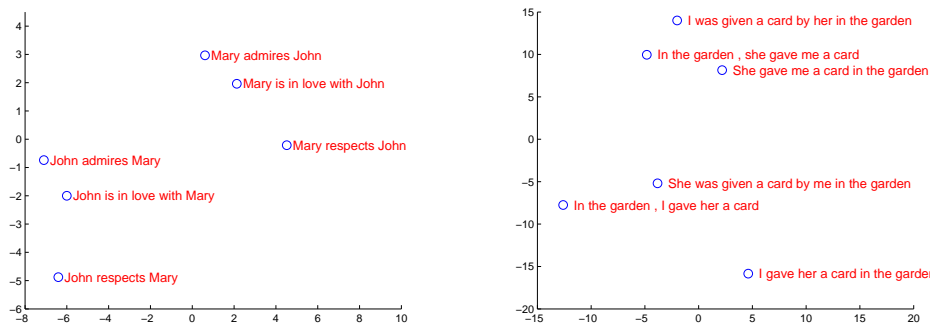

Figure 2: The figure shows a 2-dimensional PCA projection of the LSTM hidden states that are obtained after processing the phrases in the figures. The phrases are clustered by meaning, which in these examples is primarily a function of word order, which would be difficult to capture with a bag-of-words model. Notice that both clusters have similar internal structure.

One of the attractive features of our model is its ability to turn a sequence of words into a vector of fixed dimensionality. Figure 2 visualizes some of the learned representations. The figure clearly shows that the representations are sensitive to the order of words, while being fairly insensitive to the

| Type | Sentence |
|---|---|
| **Our model** | Ulrich UNK , membre du conseil d' administration du constructeur automobile Audi , affirme qu' il s' agit d' une pratique courante depuis des années pour que les téléphones portables puissent être collectés avant les réunions du conseil d' administration afin qu' ils ne soient pas utilisés comme appareils d' écoute à distance . |
| **Truth** | Ulrich Hackenberg , membre du conseil d' administration du constructeur automobile Audi , déclare que la collecte des téléphones portables avant les réunions du conseil , afin qu' ils ne puissent pas être utilisés comme appareils d' écoute à distance , est une pratique courante depuis des années . |
| **Our model** | " Les téléphones cellulaires , qui sont vraiment une question , non seulement parce qu' ils pourraient potentiellement causer des interférences avec les appareils de navigation , mais nous savons , selon la FCC , qu' ils pourraient interférer avec les tours de téléphone cellulaire lorsqu' ils sont dans l' air " , dit UNK . |
| **Truth** | " Les téléphones portables sont véritablement un problème , non seulement parce qu' ils pourraient éventuellement créer des interférences avec les instruments de navigation , mais parce que nous savons , d' après la FCC , qu' ils pourraient perturber les antennes-relais de téléphonie mobile s' ils sont utilisés à bord " , a déclaré Rosenker . |
| **Our model** | Avec la crémation , il y a un " sentiment de violence contre le corps d' un être cher " , qui sera " réduit à une pile de cendres " en très peu de temps au lieu d' un processus de décomposition " qui accompagnera les étapes du deuil " . |
| **Truth** | Il y a , avec la crémation , " une violence faite au corps aimé " , qui va être " réduit à un tas de cendres " en très peu de temps , et non après un processus de décomposition , qui " accompagnerait les phases du deuil " . |

Table 3: A few examples of long translations produced by the LSTM alongside the ground truth translations. The reader can verify that the translations are sensible using Google translate.

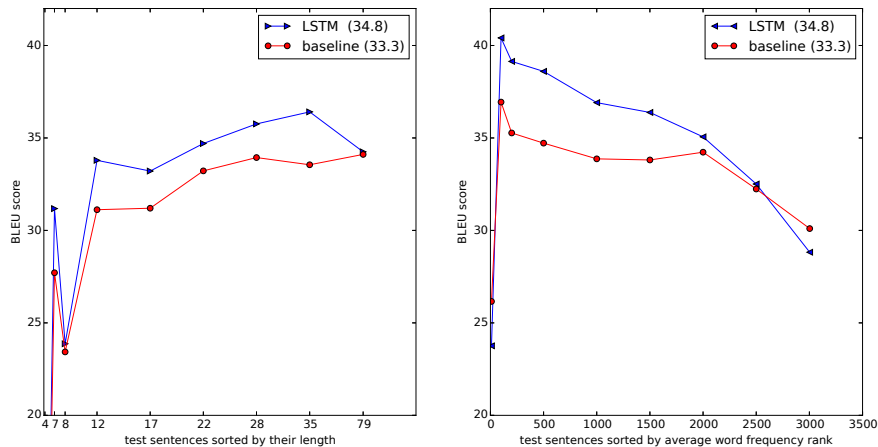

Figure 3: The left plot shows the performance of our system as a function of sentence length, where the x-axis corresponds to the test sentences sorted by their length and is marked by the actual sequence lengths. There is no degradation on sentences with less than 35 words, there is only a minor degradation on the longest sentences. The right plot shows the LSTM's performance on sentences with progressively more rare words, where the x-axis corresponds to the test sentences sorted by their "average word frequency rank".

replacement of an active voice with a passive voice. The two-dimensional projections are obtained using PCA.

## 4 Related work

There is a large body of work on applications of neural networks to machine translation. So far, the simplest and most effective way of applying an RNN-Language Model (RNNLM) [23] or a

Feedforward Neural Network Language Model (NNLM) [3] to an MT task is by rescoring the n-best lists of a strong MT baseline [22], which reliably improves translation quality.

More recently, researchers have begun to look into ways of including information about the source language into the NNLM. Examples of this work include Auli et al. [1], who combine an NNLM with a topic model of the input sentence, which improves rescoring performance. Devlin et al. [8] followed a similar approach, but they incorporated their NNLM into the decoder of an MT system and used the decoder's alignment information to provide the NNLM with the most useful words in the input sentence. Their approach was highly successful and it achieved large improvements over their baseline.

Our work is closely related to Kalchbrenner and Blunsom [18], who were the first to map the input sentence into a vector and then back to a sentence, although they map sentences to vectors using convolutional neural networks, which lose the ordering of the words. Similarly to this work, Cho et al. [5] used an LSTM-like RNN architecture to map sentences into vectors and back, although their primary focus was on integrating their neural network into an SMT system. Bahdanau et al. [2] also attempted direct translations with a neural network that used an attention mechanism to overcome the poor performance on long sentences experienced by Cho et al. [5] and achieved encouraging results. Likewise, Pouget-Abadie et al. [26] attempted to address the memory problem of Cho et al. [5] by translating pieces of the source sentence in way that produces smooth translations, which is similar to a phrase-based approach. We suspect that they could achieve similar improvements by simply training their networks on reversed source sentences.

End-to-end training is also the focus of Hermann et al. [12], whose model represents the inputs and outputs by feedforward networks, and map them to similar points in space. However, their approach cannot generate translations directly: to get a translation, they need to do a look up for closest vector in the pre-computed database of sentences, or to rescore a sentence.

## 5 Conclusion

In this work, we showed that a large deep LSTM with a limited vocabulary can outperform a standard SMT-based system whose vocabulary is unlimited on a large-scale MT task. The success of our simple LSTM-based approach on MT suggests that it should do well on many other sequence learning problems, provided they have enough training data.

We were surprised by the extent of the improvement obtained by reversing the words in the source sentences. We conclude that it is important to find a problem encoding that has the greatest number of short term dependencies, as they make the learning problem much simpler. In particular, while we were unable to train a standard RNN on the non-reversed translation problem (shown in fig. 1), we believe that a standard RNN should be easily trainable when the source sentences are reversed (although we did not verify it experimentally).

We were also surprised by the ability of the LSTM to correctly translate very long sentences. We were initially convinced that the LSTM would fail on long sentences due to its limited memory, and other researchers reported poor performance on long sentences with a model similar to ours [5, 2, 26]. And yet, LSTMs trained on the reversed dataset had little difficulty translating long sentences.

Most importantly, we demonstrated that a simple, straightforward and a relatively unoptimized approach can outperform a mature SMT system, so further work will likely lead to even greater translation accuracies. These results suggest that our approach will likely do well on other challenging sequence to sequence problems.

## 6 Acknowledgments

We thank Samy Bengio, Jeff Dean, Matthieu Devin, Geoffrey Hinton, Nal Kalchbrenner, Thang Luong, Wolfgang Macherey, Rajat Monga, Vincent Vanhoucke, Peng Xu, Wojciech Zaremba, and the Google Brain team for useful comments and discussions.

## Footnotes

[1]There several variants of the BLEU score, and each variant is defined with a perl script.

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
