[Reviews · NeurIPS 2014]

Submitted by Assigned_Reviewer_13

On the plus side: the paper has an interesting and novel idea, and I believe the experimental investigation is competent and complete.
On the minus side: I am skeptical that this idea could be a practical solution to MT, and I think your last translated-sentence example kind of shows the kind of weirdness that can result. In my mind, a solution has to be scalable in principle to long sentences, and I think it's clear that your method cannot.

Fix: 28.w
Summary: Accept but not very strongest accept.

Submitted by Assigned_Reviewer_15

This paper describes a use of LSTMs to do machine translation. The models are trained to first consume the text in the source language (word-by-word) and then to produce the text in the target language (word-by-word). At the end of the input sentence, the hidden layer(s) of the network contain a representation of the source-side sentence. This is an elegant model, and I am inclined to accept it, despite the fact that it only "works" for sentences that do not have infrequent words.

The paper does have some major holes in the experiments, which should be addressed in any revision:
1) The paper used 14 interpolated LSTMs to achieve improvements on the overall dataset with rescoring. But what happens if you do exactly the same thing, but just use the LSTM as a language model, i.e. what happens if it is not conditioned on the source-side? Past research indicates that this could give just as much improvement.
2) The sentences with infrequent words are likely to be much shorter than the others - the longer a sentence is, the more likely it is to hit an infrequent word. What is the average length of the subset without infrequent words?
3) Was the LSTM trained specifically on the common-word-only subset? What happens if the baseline system is trained on the same subset?
4) Are the poor BLEU scores for the entire test set reflective of an anomaly in BLEU (e.g. they would be good, but BLEU doesn't count synonyms), or is it really bad? Would scoring with Meteor make a difference?
Summary: This paper presents the elegant idea of translating from source to target languages with an LSTM. The experimental results are not convincing.

Submitted by Assigned_Reviewer_30

SUMMARY:
This paper introduces an LSTM-based approach to map arbitrary-length sequences to other arbitrary-length sequences, which is a required property in fully neural machine translation. The LSTM is trained to produce a fixed-size representation of a sentence, which then serves as the initial state of a second LSTM which generates a translated sentence. The authors evaluate their architecture on two machine translation datasets.

GENERAL OPINION:
The ideas represented in this work are extremely interesting, and I love the elegance and simplicity of the proposed RNN architecture. I am happy with the changes made in the second version in terms of performance.

REMARKS:

Figure 1: If I'm correct, the symbols W, X, Y, and Z are samples, and used as input for the next time step, right? If so, it might help to put arrows indicating this (output to input of next time step).

line 88: What is BLUE point and BLUE score? A short explanation or reference would help.

line 92: "a BLUE score of 28.w"

lines 112 to 118: I'm somewhat sceptical that common RNNs would find it hard to model "long-term dependencies", as word sequences are not that long at all. Most sentences are not much longer than a dozen or so words, which falls well into the dependency length that can be modelled by RNNs. I do believe RNNs would be slower and harder to train though, so please rephrase that sentence.

lines 138-139: Please refer to Alex Graves work on multi-layered LSTMs: A. Graves, A. Mohamed, G. Hinton. Speech Recognition with Deep Recurrent Neural Networks. ICASSP 2013, Vancouver, Canada.

line 154: Please remain consistent with notations. "DE-> EN" is not much shorter than "German -> English".

line 183: What is a beam search decoder? Reference? Explanation?

lines 191-193: "any translation that has an UNK cannot possibly correct". Why not? Either I'm misunderstanding something here, or this sentence is wrong. If there is an UNK in the input sentence, it makes sense that there would be one in the output, no? Conversely, if there are no UNKs in the input dataset at all, there would be no reason to include UNKs at all. Please clarify.

line 197: "Le Mans University"

line 249: "more careful tuning"

Figure 2: It would be more convincing to have some more examples. The ones presented now could be cherry-picked.

Summary: The idea of the paper is good and very interesting, providing an elegant neural solution to machine translation
Author Feedback
Author rebuttal: We thank the reviewers for their comments. Due to an error in our experiments, we trained our models on less than 25% of the available training data. We have also implemented an improved implementation that allowed us to reduce the size of the minibatch by a factor of 8 without sacrificing speed, and we doubled the size of the vocabulary of our models (so the number of parameters in a single LSTM is 380M, in which a sentence is represented with 8K scalars). As a result of these three changes, our results have improved greatly.

On the EN → FR dataset, a combination of 6 LSTMs now achieves a BLEU score of 30.77 on all sentences using direct decoding, including ones with infrequent words (for comparison, the decoding accuracy of our model was 16.2 BLEU points in the original submission). Thus, while the baseline is 33.3, our model is no longer far from it. In addition, our system surpasses the accuracy of the baseline on more than half of the sentences in the test set.

When we rerank EN → FR 1000-best lists, an ensemble of 6 LSTMs improves the baseline from 33.3 to 36.1, which represents a modest improvement (0.3 BLEU points) over state of the art on WMT’14, which is 35.8.

We agree that our reranking experiments need to be compared with a regular LSTM-LM. And while it is true that our model cannot handle long sentences easily, our analysis shows that it handles sentences of up to length 30 without loss of accuracy in terms of the BLEU score -- we will include the analysis in final version of the paper. Longer sentences can be partitioned into shorter fragments that are translated independently, and a small modification to the model can prevent the “sharp discontinuities” that occur with independent translations.

We were indeed unaware of the work "Learning Phrase Representations using RNN Encoder-Decoder for Statistical Machine Translation" [1] as it was published only a few days before the NIPS submission deadline. Having read that paper carefully we nonetheless found important differences with our work -- namely, the RNN model is trained on short phrases that were extracted from a phrase table of an standard MT system, while our model was trained directly on entire sentences. More importantly, our model achieves substantially better results: they achieve 34.54 while we achieve 36.1 (by rescoring the aforementioned 33.3 system). Much more significant is the fact that our model approaches baseline accuracy completely on its own, using direct left-to-right decoding, which is not the case for [1].

We have tried common RNNs and found that it was difficult to get them working -- at the very least, they didn’t work right away. Perhaps the common RNN would have worked with more tuning effort, but the LSTM worked on our first try and required little parameter tuning. Note also that the work of [1] also did not use common RNNs for their phrase to phrase system, as they introduced architectural changes to their RNN that make very similar to the LSTM.